# Characterization of Operational Vibrations of Steel-Girder Highway Bridges via LiDAR

Adriana Trias-Blanco [1], Jie Gong [2] and Franklin L. Moon [2,*]

1   Department of Civil and Environmental Engineering 1, Rowan University, Glassboro, NJ 08028, USA
2   Department of Civil and Environmental Engineering, Rutgers University, Piscataway, NJ 08854, USA
*   Correspondence: franklin.moon@rutgers.edu

**Abstract:** This research is motivated by the need for rapidly deployable technologies such as wireless, non-contact or remote sensing for evaluating bridges under operating conditions to minimize the data collection time, avoid the disruption of traffic and increase the inspector's safety. The objective established for this research is to explore the use of remote sensing (e.g., Light Detection and Ranging (LiDAR)) for characterizing the structural vibration of bridges to support and improve bridge assessment practices. To satisfy this objective, a field study was performed on a 12-span steel stringer bridge in the Philadelphia region. This structure was subjected to extensive LiDAR scanning and conventional vibration data collection through the use of accelerometers for validation purposes. The analysis of the data collected in the field revealed LiDAR's capability for detecting the structure's vibration. The field data displayed an error for LiDAR vs. accelerometers of between 1.9 and 10 percent. Additionally, numerical modeling was performed on MATLAB to allow for a better understanding of the interaction between the scanner and the structure. The numerical model presents a vibrating plate to represent a simply supported single-span bridge and a terrestrial LiDAR sensor located under the plate which scans while it is vibrating constantly without attenuation. Finally, a set of recommendations were established for the use of LiDAR scanning to evaluate the structure's frequency of vibration.

**Keywords:** structural vibration; LiDAR; point cloud; remote sensing; bridge engineering

## 1. Introduction

According to the American Society of Civil Engineers' (ASCE) Infrastructure Report Card [1], close to 42% of all bridges in the United States are at least 50 years old (even though they were designed for a 50-year service life); 7.5% are structurally deficient, which indicates that they are very close to the end of their life-cycle unless significant interventions are undertaken. Although progress is being made to reduce the number of structurally deficient bridges (as evidenced by a 2.5% decrease in four years), whether or not this progress can be maintained or accelerated remains an open question. Given the sheer number of bridges older than 50 years, as well as the ones approaching this age, it is clear that we cannot simply hope to build ourselves out of this challenge by the wholesale replacement of all old and deficient bridges. Rather, there is a need to develop rapid and efficient diagnosis, prognosis and repair techniques to safely and cost-effectively extend the life-cycle of our current bridge stock. Although all these tools are needed, assessment is arguably the most critical. Without the ability to identify and characterize deficiencies at their early stages, prognosis and various repair strategies simply cannot be brought to bear effectively. For this reason, this research intends to provide recommendations for the implementation of LiDAR on a TLS platform.

Over the last decade, bridge owners have begun to explore augmenting conventional assessment approaches with nondestructive evaluation (NDE) [2] and structural health monitoring (SHM) technologies [3]. The primary barrier associated with the use of these

technologies is not simply cost but rather the disruption to the traveling public which generally accompanies their implementation [4]. As a result, rapidly deployable technologies such as wireless, non-contact or remote sensing are important technologies for the future of SHM.

This growing interest in sensing technologies that may be deployed in a rapid or noninvasive manner has focused on the identification of application scenarios that demonstrate the value of such technologies over conventional approaches. Towards that end, the research presented herein aims to explore the use of remote sensing (e.g., Light Detection and Ranging (LiDAR)) to support and improve bridge assessment. LiDAR sensors have the ability to capture dense point clouds that define the geometry of objects in a remote (non-contract) manner. For this research, the team used a FARO Focus S 150, with a vertical range of 300°, a horizontal range of 360°, and a minimum step size (angular distance between points) of 0.009° (1.57E-4 rad). The scanner has a range error of ±1 mm [5]. The sampling frequency was set to 244 points/sec, at a resolution of 0.035° (four times the minimum step size). Figure 1 shows the principal of operations of a typical LiDAR sensor.

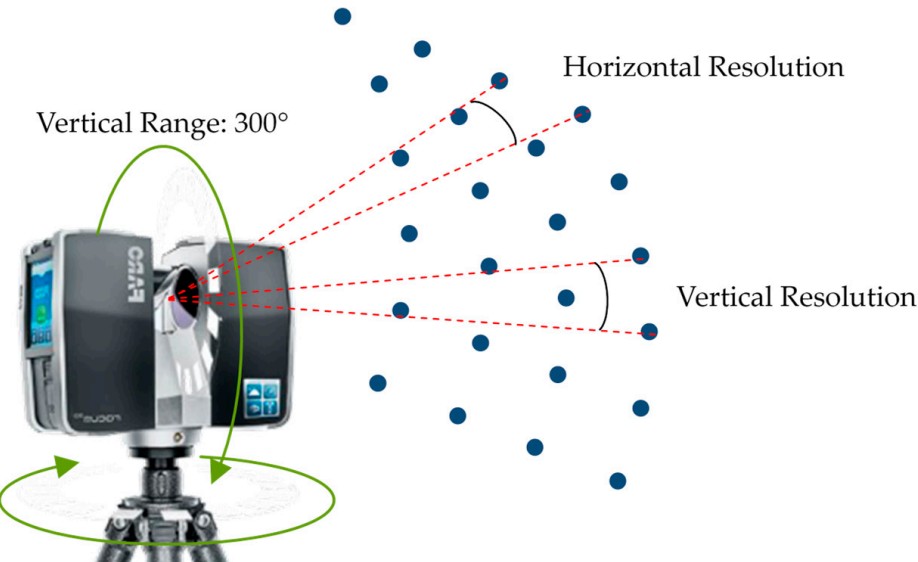

**Figure 1.** Principals of Operations of LiDAR.

LiDAR sensors may be deployed as Terrestrial Laser Scanners (TLS), Airborne Laser Scanners (ALS) or Mobile Laser Scanners (MLS) [6]. Over the last few decades, these sensors have demonstrated an ability to efficiently and rapidly gather a large amount of high-quality spatial data, surpassing the data collection capacity of the theodolite and total station [7]. One area that has attracted attention is the use of LiDAR to support structural assessment activities. The key benefits of LiDAR over conventional approaches in this application area include (a) a more complete and quantitative record of geometry that may be used for future "change detection" activities [8], (b) providing a platform to better conceptualize the structure by exploring the point cloud [7], (c) reducing the potential for documentation errors [9], (d) helping to standardize the inspection process and reduce the reliance on qualitative and subjective information [10] and (e) providing full-field geometry/displacement information, which contextualizes the information for more complete data interpretation. Moreover, LiDAR has improved the quality of bridge monitoring by allowing one to measure bridge displacements in a contactless manner [11]. In the field of vibration monitoring, traditional methods implement the use of accelerometers to capture the behavior of structures under (a) ambient vibrations or (b) imposed loading. This information is then used to calibrate computational models that allow one to study and determine the structural capacity, condition stage and structural identification [12,13].

Laser-based approaches for measuring structural vibrations are currently under exploration, as researchers investigate the detection of passing vehicles through the implementation of laser doppler vibrometers [14], finite element model calibration by measuring deflection and mode shapes by the interpretation of data captured via LiDAR and laser doppler vibrometers [15].

It is important to establish a comparison between conventional vibration sensors (accelerometers) and TLS. Accelerometers present demonstrated benefits and shortcomings for measuring the vibration responses of structures. Some of the benefits are: (a) the high sampling rate [16], (b) the low cost compared to that of TLS [17] and (c) the lower amount of data compared to that of TLS, while some of the shortcomings are: (a) the single-point data collection, while LiDAR scanners are able to gather full-field data, (b) the fact that installation requires traffic disruption [18] and (c) the fact that it can only be used to measure vibration, while TLS data (point cloud) also provide the geometry of the elements [19].

## 2. Objectives and Approach

The overarching aim of the research is to explore the use of remote sensing—more specifically, LiDAR—as a contributor to the vibration analysis of bridges. To achieve this goal, this research leveraged a diverse set of approaches that included the use of data collected from (a) operating highway bridges and (b) a mathematical model used to simulate the dynamic response of typical highway bridges. These data collection efforts were supported by various data analysis techniques, data visualization and interpretation approaches and numerical simulation modeling.

The field work designated for the comprehension of the effects of response vibrations regarding the point cloud data required the use of a numerical model that served as a reference, where different variables could be evaluated. The variables tested in this controlled digital environment were (1) scanner-dependent (Resolution, Temporal frequency, Angular frequency) and (2) structure-dependent (Amplitude of vibration, Frequency of vibration).

## 3. Field Data Collection

The scanned structure is a steel multi-girder bridge with a total of 11 spans with a combination of single- and double-spans. The girders are braced laterally through X- and K-shaped frames and are supported by concrete piers. The structure was built in 1986, utilizing the substructure, the foundation and the preexisting bridge built in 1956. The bridge was reported to have excessive vibration, hence the need for this research project.

During the analysis of the bridge point cloud data captured via the LiDAR of an operating bridge, localized ripples were noted in horizontal elements that were expected to be planar in nature (such as steel girder flanges). Figure 2 shows an example of the ripples observed. These ripples were measured to have an amplitude of approximately 25 mm (25 times the range error of the scanner) and were more significantly detected within a range of 4 m around the scanner. It was hypothesized that these ripples are a result of the bridge vibrating under truck traffic. If true, given the broadband nature of traffic excitation (1–5 Hz for typical highway bridges [20–22]), analyzing these ripples should be able to provide information about the dynamic properties of the test structure. That is, their frequency is likely more related to the dynamic properties of the structure than the frequency of the loading (since it is broadband within the frequency range of interest). This paper presents the results obtained after the examining this hypothesis through the use of data obtained from an operating highway bridge, along with the use of numerical and physical models. The following sections outline the research conducted using each of these approaches.

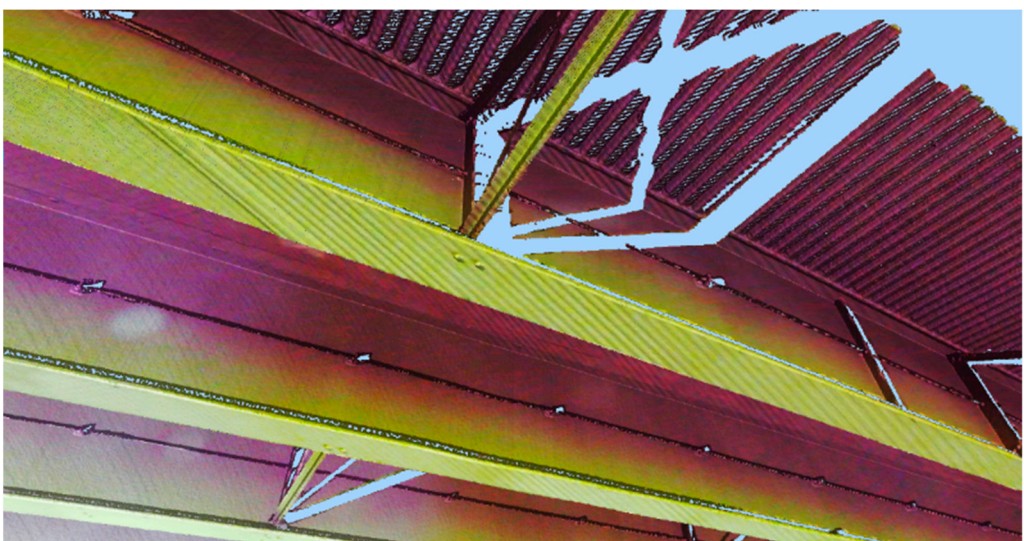

**Figure 2.** Ripple pattern found in the point cloud of the collected field data caused by operating traffic.

### 3.1. Observation from an Operating Highway Bridge

This ripple phenomenon was easily observed throughout the point cloud data, as shown before, indicating that this was not a one-time event. Then, the focus oriented toward determining the potential extraction of modal frequencies from the point cloud data as a means of characterizing the dynamic performance of the bridge. This subsection presents how the Temporal Frequency and Angular Frequency of the point cloud data were calculated and how they were used to estimate the frequency of bridge vibrations.

### 3.1.1. Temporal Frequency during the Field Data Collection

The temporal frequency of the LiDAR data is defined as the inverse of the time it takes for two horizontally consecutive data points to be registered. Figure 3 explains this concept, where points E and H are considered to be horizontally consecutive points since they have the same vertical rotational angle and were taken in two consecutive LiDAR revolutions. Similarly, points A-D, B-E and F-I are also horizontally consecutive points. Then, the inverse of the time between these two points is the Temporal Frequency ($f_t$), as shown in Equation (1), where $t_p$ is the time between two consecutive points. During the field data collection, the scanner was set to perform at a $f_t$ of 24Hz; therefore, the $t_p$ was 0.04167 s.

$$f_t = \frac{1}{t_p},\tag{1}$$

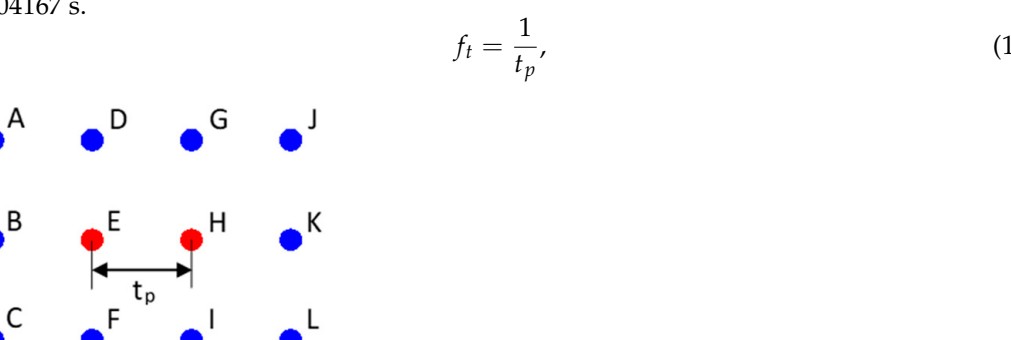

**Figure 3.** Temporal frequency of the LiDAR data.

### 3.1.2. Angular Frequency during Field Data Collection

The angular frequency of the LiDAR data is defined as the horizontal degrees covered per second by the scanner. The Angular Frequency ($f_a$) can be found by multiplying the temporal frequency of the scanner by the step size or resolution (the angular distance between two consecutive points), as shown in Equation (2):

$$f_a = (f_t - 1) \times Resolution \tag{2}$$

In Equation (2), since the temporal frequency is the number of points per second, one unit must be subtracted, since the equation seeks the number of spaces between the data points. The final units of Fa are degrees per second. During the field data collection, the scanner was set to perform at a resolution of 0.035°, resulting in an $f_a$ of 0.805 deg/s.

3.1.3. Measurement of the Bridge Frequency of Vibrations

Based on the setting of the scanner at the moment of the field data collection, the bridge frequency of vibration was calculated through the following process:

- Calculate the angular distance ($Dist_{ang}$) between the two low points (of a ripple), considering the LiDAR's location following Equation (3). *X* and *Y* are the coordinates of the selected data points.

$$Dist_{ang} = \cos^{-1}\left(\frac{X_1 \cdot X_2 + Y_1 \cdot Y_2}{\sqrt{X_1^2 + Y_1^2} \cdot \sqrt{X_2^2 + Y_2^2}}\right) \tag{3}$$

- Calculate the period between ripples ($T_R$), which is the time, in seconds, between the occurrence of two consecutive low points, following Equation (4).

$$T_R = \frac{Dist_{ang}}{(f_t - 1) \times Resolution} = \frac{Dist_{ang}}{f_a} \tag{4}$$

- Calculate the bridge frequency of vibration ($f_o$), following Equation (5).

$$f_o = \frac{1}{T_R} \tag{5}$$

Random samples of five or more consecutive ripples were taken to measure the vibrating frequency of spans 02, 03, 05 and 07. The previously explained steps and equations were applied to measure the frequency of vibration of each span. LiDAR's parameters during the data collection are summarized in Table 1.

**Table 1.** LiDAR's parameters during the field data collection.

| Resolution (°) | Temporal Frequency $f_t$ (Points/s) | Angular Frequency $f_a$ (Degrees/s) |
| --- | --- | --- |
| 0.035 | 24 | 0.805 |

## 4. Field Data Analysis and Results

To verify the findings obtained through the LiDAR data analysis, the results were compared to a full dynamic test performed for the bridge, where different mode shapes with the correspondent frequency of vibration were found. The dynamic test consisted of the installation of accelerometers to measure the bridge vibrating response on key locations, as shown in Figure 4. The spans included within this particular test were 2, 3, 4, 7 and 8; therefore, these were the only spans where the LiDAR data could be verified. The shape of Mode 1 and the correspondent frequency of vibration found from the data analysis of these spans are presented in Figure 5.

The results gathered from the analysis of the field data are summarized in Table 2; this table includes the results gathered from the dynamic test. The Angular Distance, Ripple Period and Frequency of Vibration from LiDAR include a range corresponding to the 95% confidence interval, while the percent error calculated for LiDAR's Frequency of Vibration follows Equation (6). Additionally, the natural frequencies for the first eight modes of spans 02, 03–04 and 07–08 are listed in Table 3.

$$\%error(Frequency) = \frac{(Frequency\ LiDAR\ data - Frequency\ Dynamic\ test)}{Frequency\ Dynamic\ test} \tag{6}$$

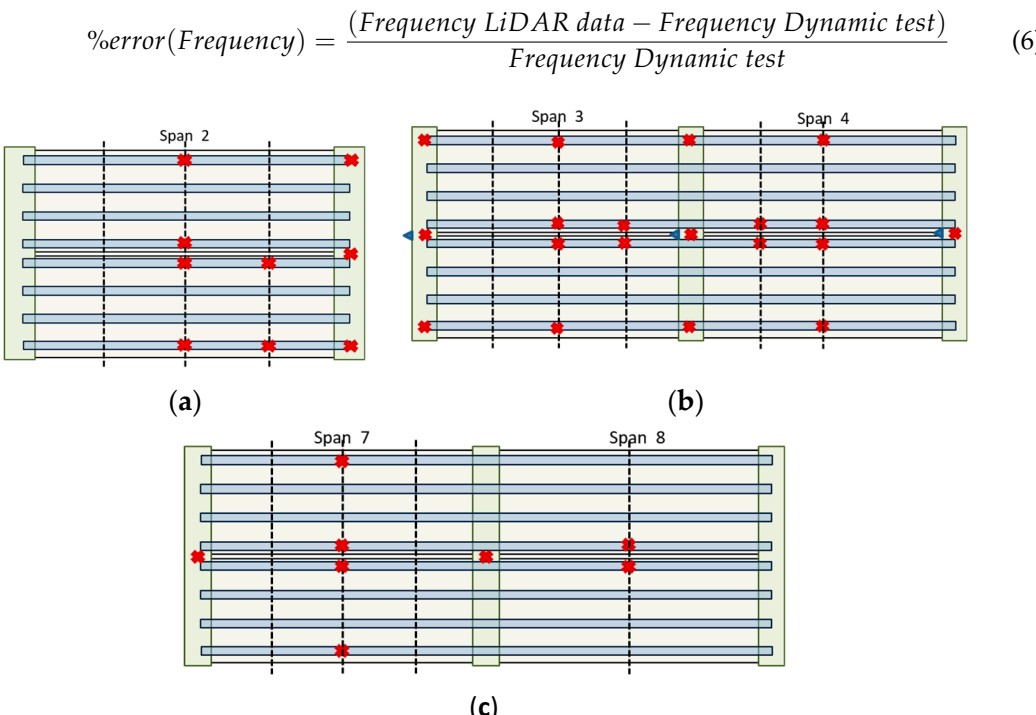

**Figure 4.** Instrumentation Layout: (**a**) Span 2; (**b**) Spans 3 and 4; (**c**) Spans 7 and 8.

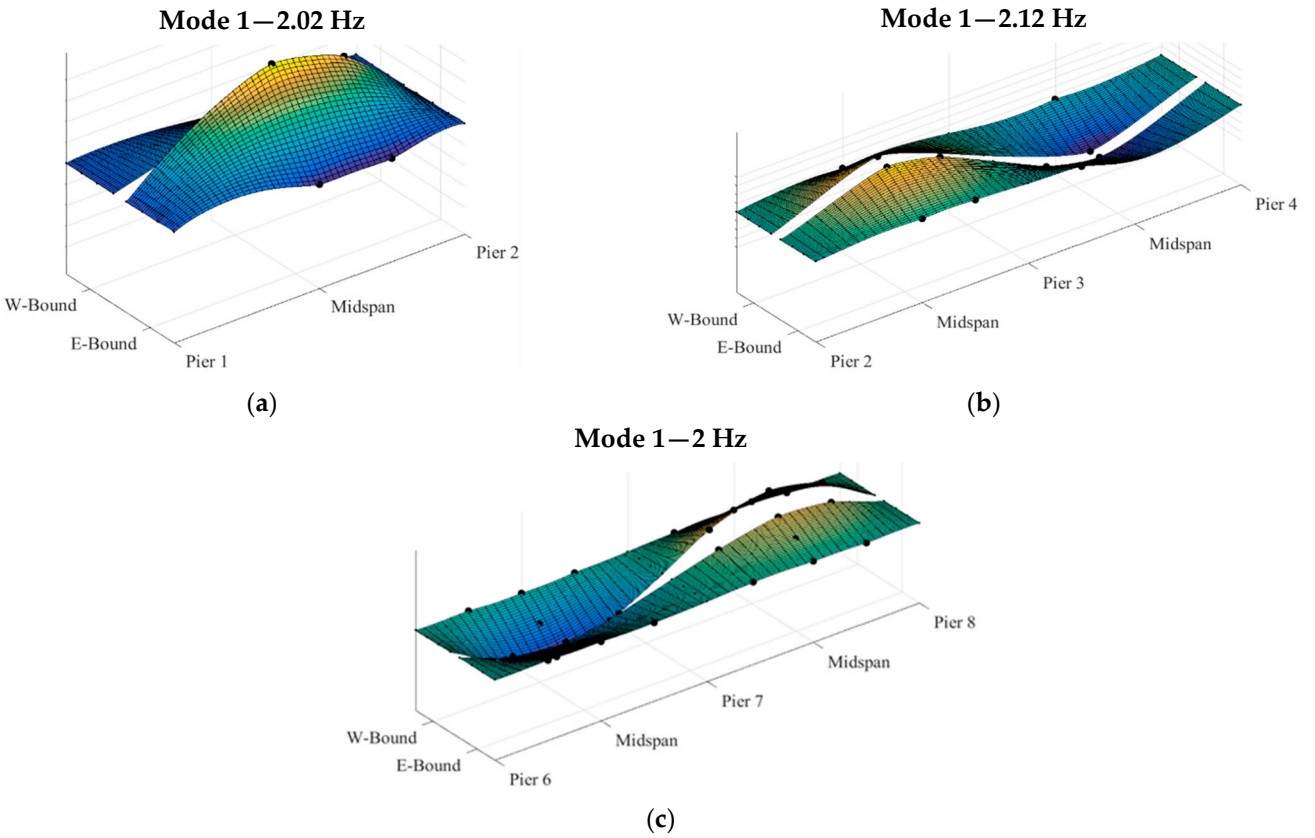

**Figure 5.** Shape of Mode 1 and the correspondent frequency from the dynamic testing (accelerometers): (**a**) Span 2; (**b**) Spans 3 and 4; (**c**) Spans 7 and 8 (not to scale).

**Table 2.** Bridge Frequency of Vibration.

| Span Number | Average Angular Distance $Dist_{ang}$ (°) | Standard Deviation $Dist_{ang}$ (°) | Ripple Period $T_R$ (s) | Frequency of Vibration $f_o$ from the LiDAR Data (Hz) | Frequency of Vibration $f_o$ from the Dynamic Test (Hz) |
|---|---|---|---|---|---|
| 2 | 0.42 (±0.03) | 0.03 | 0.51 (±0.04) | 1.98 ± 0.13 (−1.98%) | 2.02 |
| 3 and 4 | 0.35 (±0.06) | 0.04 | 0.43 (±0.08) | 2.35 ± 0.42 (10.8%) | 2.12 |
| 7 | 0.40 (±0.09) | 0.05 | 0.48 (±0.10) | 2.07 ± 0.49 (3.5%) | 2.00 |
| 8 | 0.36 (±0.03) | 0.01 | 0.43 (±0.04) | 2.30 ± 0.22 (15.0%) | 2.00 |

**Table 3.** Natural Frequencies of the first eight modes of vibration for Spans 2, 3–4 and 7–8 (accelerometers data).

| Mode Shape | Span 2 | Span 3–4 | Span 7–8 |
|---|---|---|---|
| 1 | 2.02 | 2.12 | 2.00 |
| 2 | 2.12 | 2.47 | 2.03 |
| 3 | 2.62 | 2.92 | 2.10 |
| 4 | 2.72 | 3.53 | 2.44 |
| 5 | 3.13 | 5.04 | 2.51 |
| 6 | 3.88 | 5.55 | 2.54 |
| 7 | 4.29 | 5.85 | 2.83 |
| 8 | 5.04 | 7.26 | 2.93 |

## 5. Numerical Modeling

This section presents the different parameters established to define the numerical model developed in order to further examine the findings from the field data described above. The goal of this model was to simulate the bridge/scanner interaction and identify the influence of various structural parameters (e.g., fundamental frequency, vibration amplitude) and data acquisition parameters (e.g., temporal and angular frequency, distance to target) on the ability of LiDAR data to be used to estimate the fundamental frequency of a structure. The code written to satisfy the objectives of this section considered a vibrating rectangular plate with the parameters shown in Table 4.

**Table 4.** Numerical model input parameters. Plate specifications.

| Parameter Name | Description | Symbol | Unit |
|---|---|---|---|
| Length | Length of the span being simulated | L | cm |
| Width | Width of the typical girder bottom flange | b | cm |
| Stiffness | Flexural Stiffness of the bridge | K | kg/cm |
| Mass | Total mass of the bridge | m | kg |
| Acceleration multiplier | Acceleration multiplier to increase the amplitude of vibration | U | Unitless |

Based on the parameters of Table 4, the natural frequency of the plate was calculated following Equation (7), where *K* is the bridge stiffness, and m is the total mass of the bridge.

$$Wn = \sqrt[2]{\frac{K}{m}} \tag{7}$$

In addition, Table 5 shows the data acquisition parameters simulated by the numerical model. Figure 6 indicates the global sign convention adopted in the code.

**Table 5.** Numeral model input parameters. Scanner settings.

| Parameter Name | Description | Symbol | Unit |
|---|---|---|---|
| Resolution | Angular distance between two consecutive points. | Deg | Degrees |
| Distance to Target | Perpendicular distance between the scanner and plate | Len | cm |
| Temporal Frequency | Number of scanner revolutions per second | ft | rev/s |
| Scanner Longitudinal Location | Location of the scanner with respect to the longitudinal direction of the plate | LongLoc | cm |
| Scanner Transverse Location | Location of the scanner with respect to the transverse direction of the plate | TranLoc | cm |

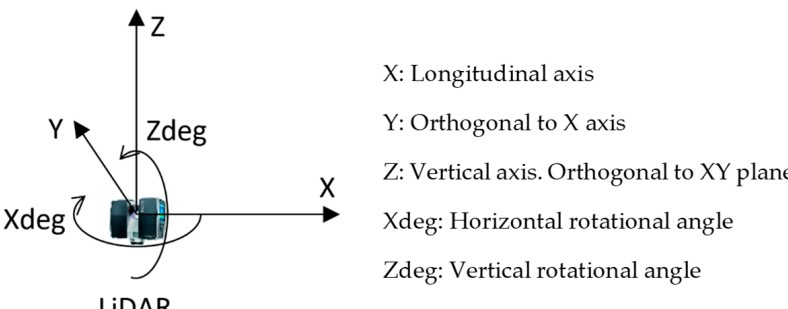

X: Longitudinal axis

Y: Orthogonal to X axis

Z: Vertical axis. Orthogonal to XY plane

Xdeg: Horizontal rotational angle

Zdeg: Vertical rotational angle

**Figure 6.** Global Sign Convention.

Once all parameters were determined, the algorithm of the numerical model was applied as follows:

- Calculate the Xdeg of each point with respect to the scanner and assign a distance *alpha*. Alpha is the longitudinal location of the point within the plate (see Equation (8)).

$$alpha = \tan(Xdeg) \times Len \qquad (8)$$

- Calculate the Zdeg of each point with respect to the scanner and assign a distance *beta*. Beta is the transverse distance of the point within the plate (see Equation (9)).

$$beta = \tan(Ydeg) \times Len \qquad (9)$$

- Assign an ID to each point of the point cloud, based on the scanner rotating sequence, assuming the scanner rotates forward and clockwise. The point ID (pointID) was assigned based on the following code:

```
zcover = 360; %Vertical degrees covered by the scanner.
ptsCicle = round((zcover/(Deg)),0); %Number of points of scanner revolution.
for i = 1:size(beta,1)
for j = 1:size(beta,2)
pointID(i,j) = (i − 1)+ptsCicle*(j − 1);
end
end
```

- Finally, the algorithm needed a shape function (Equation (10)), a time assigned to each point (Equation (11)) and the natural frequency that was previously calculated to calculate the vertical displacement for each point, as presented in Equation (12).

$$shape = \sin\left(\pi \times \frac{(alpha + LongLoc)}{L}\right) \qquad (10)$$

$$time = \left(\frac{pointID}{ptsCicle}\right) \times \left(\frac{1}{Rev}\right) \qquad (11)$$

$$u = U \times shape \times \sin\left(\frac{\pi \times time}{1/wn}\right) \tag{12}$$

For the initial simulations, the parameters shown in Tables 4 and 5 with the aforementioned equations were employed. The modification of the scanner location allowed for the identification of the shifts of the ripple patterns with respect to this variable and highlighted the influence of the incidence angle on the data resolution. Table 6 shows the resulting ripple patterns for two different temporal frequencies (97 and 43 Hz) and three different resolutions (0.009°, 0.09° and 0.9°). Is important to highlight that, in this initial simulation, the distance to the target (Len) was kept constant; further analysis was conducted to evaluate the influence of this parameter.

**Table 6.** Variation in the ripples pattern depending on the laser scanner configuration.

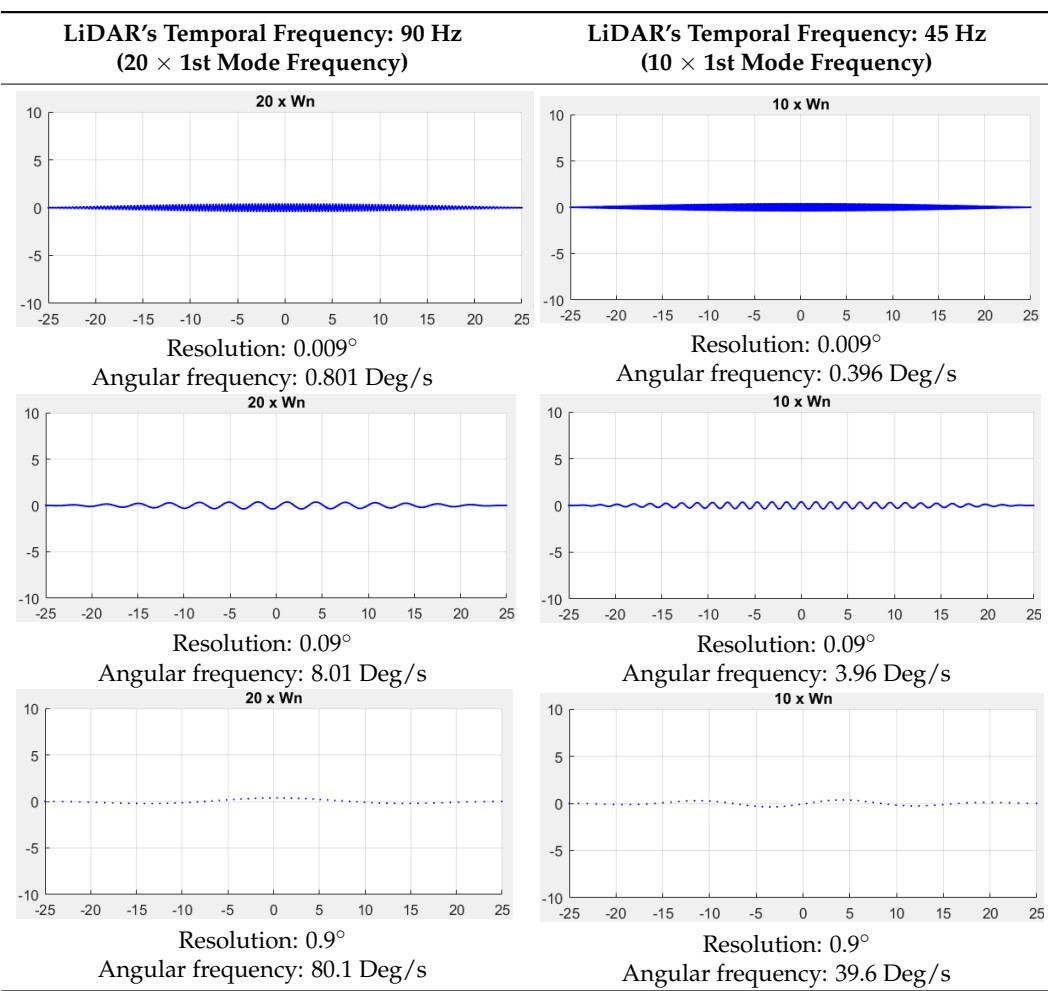

Since the angular frequency depends on the temporal frequency and resolution of the scanner, two trends can be observed in Table 6:

(1)  As the resolution increases (the angular distance between two consecutive points), there will be an increase in the angular frequency. This will increase the distance between ripples for a given vibration frequency.

(2)  As the temporal frequency decreases, there will be a decrease in the angular frequency. This will decrease the distance between ripples for a given vibration frequency.

The first observation is somewhat intuitive, but the second observation requires some explanation. By slowing down the temporal frequency while keeping the resolution constant, the scanner takes more time to cover the same area of the plate, giving the

plate more time to change position in between samples. This is equivalent to keeping the temporal frequency constant and increasing the frequency of vibration of the plate.

The distance between the scanner and the plate will also affect the characterization of the ripples. A 0.9° resolution at 9.14 m. Len has the same ripple pattern as a 0.09° resolution at 91.4 m. Len and a 0.009° resolution at 914 m. Len. All of these combinations result in an angular frequency of 26.9 Deg/s. On the other hand, Table 7 demonstrates how variating the distance to the target, without changing the resolution, can affect the ripple pattern. The blue line represents a resolution of 0.009° at a 9.14 m. distance from the scanner to the vibrating plate; the red line is the result of maintaining the same resolution and increasing the distance to 91.4 m.; finally, the black line represents a ripple pattern of a scan taken again at 0.009° while increasing the distance to 914 m.

**Table 7.** Influence of the distance to the target on the ripple pattern.

| Parameters | Ripple Pattern |
| --- | --- |
| Resolution: 0.009°<br>$f_a$: 26.1<br>Len: 9.14 m. | 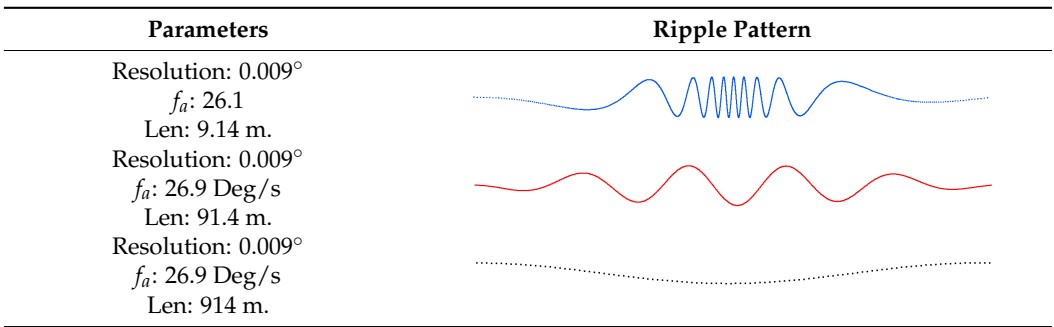 |
| Resolution: 0.009°<br>$f_a$: 26.9 Deg/s<br>Len: 91.4 m. | |
| Resolution: 0.009°<br>$f_a$: 26.9 Deg/s<br>Len: 914 m. | |

While the first two cases clearly show ripple patterns, the third case, which had the largest distance between the scanner and the plate, produced poorly defined ripples that could easily be mistaken for the actual shape of the plate. In practice, this could be erroneously interpreted as structural distortion, which could lead to false alarms related to performance concerns. This is particularly relevant for signature bridges that are generally scanned from considerable distances (in many cases, from shorelines) and have relatively large amplitude vibrations at relatively low frequencies (<0.5 Hz).

Another factor that influences the shape of the ripples is the location of the scanner, since this will determine the incidence angle between the beam of light projected from the scanner and the object scanned. Figure 7 shows a comparison of the ripple patterns captured from a scanner in three different locations: (a) under the left support, (b) under the 1/3-span of the plate and (c) under the mid-span.

As mentioned before, in this case, the incidence angle is the parameter altering the ripple pattern. This effect is similar to what was shown from the distance to the target. The incidence angle is measured, as shown in Figure 8; therefore, the smaller the incidence angle, the greater the distance between the ripples (because this corresponds to the largest distance (on the structure being scanned) between the data from subsequent passes).

For operational vibrations that have sufficient amplitudes to be captured by the LiDAR sensor, the resulting scan will display ripple patterns. By estimating the distance between the peak of these ripples (and translating this into time using the data acquisition metrics), the frequency of the vibrating object can be estimated. For the operating bridge employed in this research, the natural frequencies were estimated within 1.9% to 10%.

The frequency of vibration of the object is inversely related to the distance between the ripples (i.e., the period). On the contrary, the amplitude of vibration of the object is directly related to the amplitude of the ripples. These trends influence the manner in which ripple patterns manifest within the data, with (a) high-frequency (>10 Hz) vibration generally showing up as increased noise, (b) typical highway bridge frequencies (2–5 Hz) showing up as a clear ripple pattern spaced from 1.3° to 3.3° and (c) long-span bridges with a very low frequency (<0.3 Hz) resulting in a situation where a user may fail to recognize the presence

of ripple patterns and treat the resulting data as representative of the static geometry of the elements.

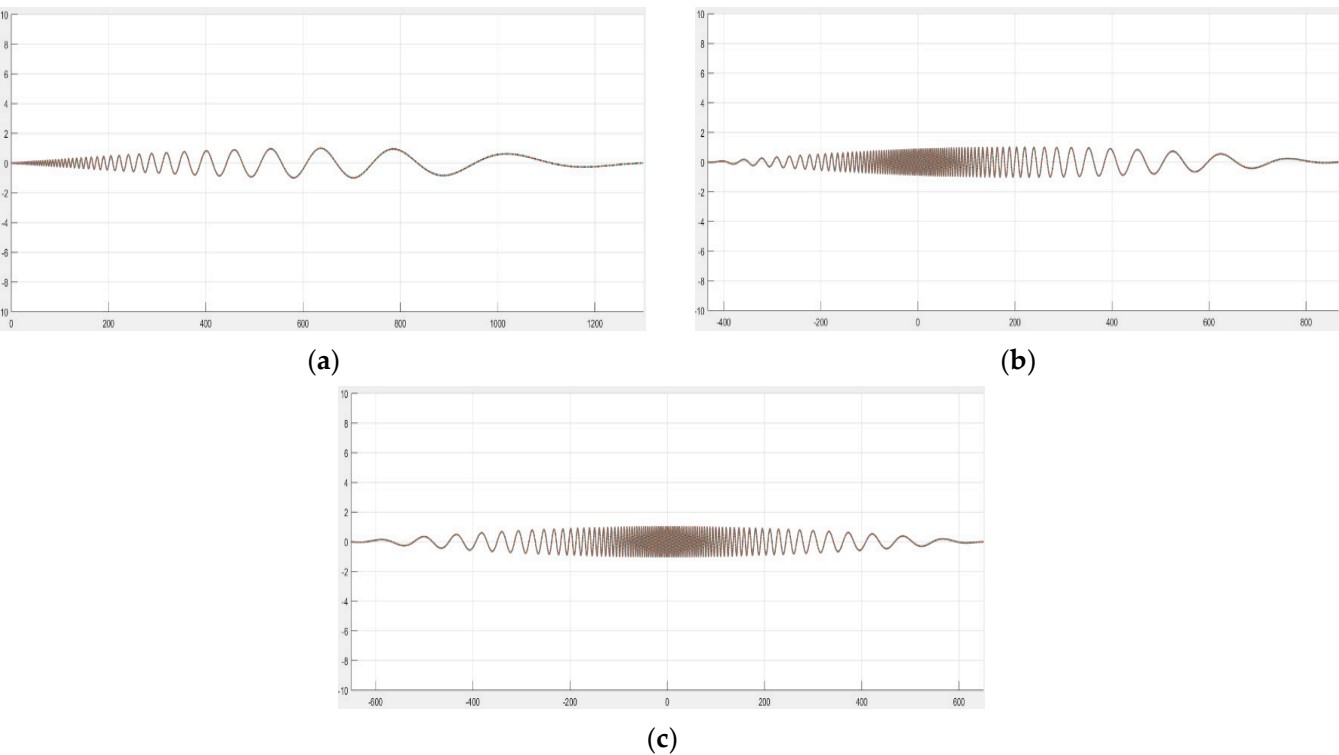

(a)

(b)

(c)

**Figure 7.** Influence of the laser scanner location on the ripples' pattern. (**a**) Scanner located underneath the left support; (**b**) Scanner located at one-third from the left support; (**c**) Scanner located at the mid−span.

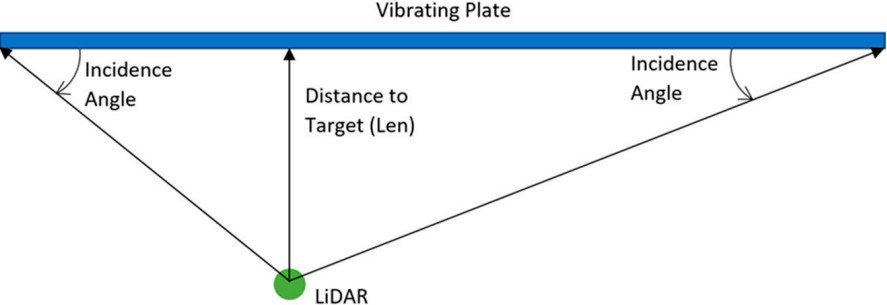

**Figure 8.** Measurement of the incidence angle.

The spacing of the ripples (Angular Distance) is directly proportional to the angular frequency ($f$a). For example, for a given vibration frequency, a scanner set to a lower angular frequency will produce ripples with narrower spacing compared to a scanner set to a higher angular frequency, which will produce ripples with broader spacing, as seen in Figure 9.

The spacing of the ripples (Angular Distance) is inversely proportional to the vibration frequency ($f_o$). For example, for a given angular frequency, if the structure is vibrating at a low frequency, the ripples will have broad spacing, while, if the structure is vibrating at a high frequency, the ripples produced will have narrow spacing, as seen in Figure 10.

$$f_o = 1.5 \text{ Hz}$$

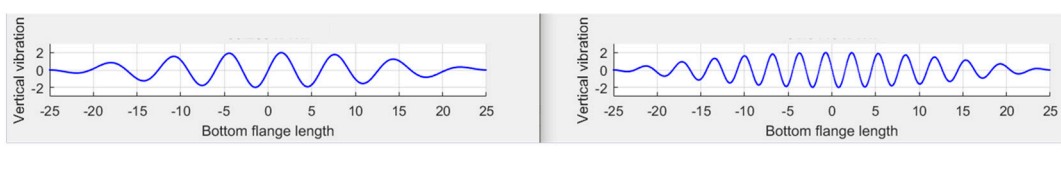

$$f_a = 90 \text{ Hz} \qquad\qquad f_a = 45 \text{ Hz}$$

**Figure 9.** Angular Distance vs. Angular Frequency.

$$f_a = 90 \text{Hz}$$

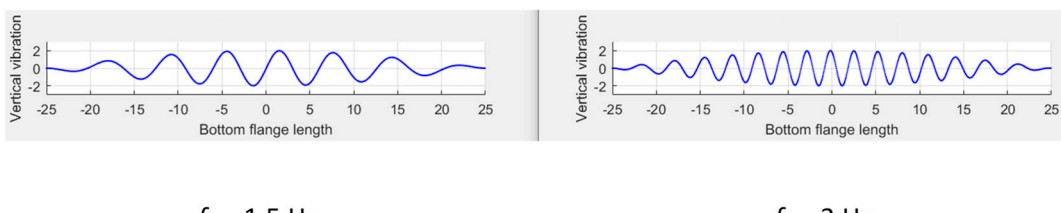

$$f_o = 1.5 \text{ Hz} \qquad\qquad f_o = 3 \text{ Hz}$$

**Figure 10.** Angular Distance vs. Vibration Frequency.

## 6. Conclusions

If the scanner's temporal frequency compared to the object's vibration frequency is sufficiently small, aliasing of the ripple pattern occurs, and the shape of the point cloud could be misinterpreted as a deformation of the scanned element. To avoid this, the temporal frequency should be kept at at least eight times the vibration frequency of the object. Therefore, the Nyquist frequency will be at least four times the object's vibration frequency.

If the angular frequency of the scanner is sufficiently high (i.e., the scanner passes over the structure at a fast rate), aliasing of the ripple pattern occurs. To avoid this, and to better characterize the ripple pattern, it is recommended that the angular frequency be selected to allow the structure to complete ten vibration cycles during scanning.

Adding to the evidence gained from this research on LiDAR's capability of capturing the structure's vibration for model analysis, it is important to highlight the effects that structural vibrations have on the acquired point cloud data. Ignoring these effects has the potential of misestimating the member's dimensions, deflection and/or local damage. It is important to identify and isolate this source of error.

The following recommendations aim to establish the minimum parameters that should be considered to ensure the characterization of the ripple patterns that result from the interaction between the vibrating structure (bridge) and the static LiDAR. For the use of LiDAR as a tool to estimate the natural frequency of a structure, the primary requirement is the proper characterization of the ripple patterns that result from vibration. To design an effective scanning protocol, it is necessary to begin by estimating both the amplitude and the frequency of the vibration that will be quantified. To choose the scanner's parameters before the data acquisition, the following three steps are recommended:

(1) Step 1—Select the maximum distance between the scanner and the object being scanned from Table 8 (reproduced below), ensuring that the amplitude of vibration is at least 1.5 times the manufacturer-specified accuracy.

(2) Step 2—Select the angular frequency to be between 3.5 and 13 degrees/s and then use Table 9 to select the desired temporal frequency to ensure no aliasing will occur. The criteria to avoid aliasing (as mentioned in the conclusions) were taken as a minimum of eight times the frequency of the vibrating object.

(3) Step 3—Based on the results from Steps 1 and 2, use Table 10 to estimate the amount of time per scan, based on the initial Step Size and Temporal Frequency.

**Table 8.** Minimum Amplitude of Vibration Detected. Local Accuracy (cm).

| | | **LiDAR's Step Size (Degrees)** | | | | | | | | |
| | | **0.009** | **0.018** | **0.035** | **0.040** | **0.070** | **0.088** | **0.140** | **0.175** | **0.280** |
|---|---|---|---|---|---|---|---|---|---|---|
| | 1 | 0.028 | 0.057 | 0.110 | 0.126 | 0.220 | 0.276 | 0.440 | 0.550 | 0.880 |
| | 2 | 0.057 | 0.113 | 0.220 | 0.251 | 0.440 | 0.553 | 0.880 | 1.100 | 1.759 |
| | 3 | 0.085 | 0.170 | 0.330 | 0.377 | 0.660 | 0.829 | 1.319 | 1.649 | 2.639 |
| | 4 | 0.113 | 0.226 | 0.440 | 0.503 | 0.880 | 1.106 | 1.759 | 2.199 | 3.519 |
| | 5 | 0.141 | 0.283 | 0.550 | 0.628 | 1.100 | 1.382 | 2.199 | 2.749 | 4.398 |
| | 6 | 0.170 | 0.339 | 0.660 | 0.754 | 1.319 | 1.659 | 2.639 | 3.299 | 5.278 |
| | 7 | 0.198 | 0.396 | 0.770 | 0.880 | 1.539 | 1.935 | 3.079 | 3.848 | 6.158 |
| | 8 | 0.226 | 0.452 | 0.880 | 1.005 | 1.759 | 2.212 | 3.519 | 4.398 | 7.037 |
| | 9 | 0.254 | 0.509 | 0.990 | 1.131 | 1.979 | 2.488 | 3.958 | 4.948 | 7.917 |
| Distance to Target (meters) | 10 | 0.283 | 0.565 | 1.100 | 1.257 | 2.199 | 2.765 | 4.398 | 5.498 | 8.797 |
| | 20 | 0.565 | 1.131 | 2.199 | 2.513 | 4.398 | 5.529 | 8.796 | 10.99 | 17.59 |
| | 30 | 0.848 | 1.696 | 3.299 | 3.770 | 6.597 | 8.294 | 13.19 | 16.49 | 26.39 |
| | 40 | 1.131 | 2.262 | 4.398 | 5.027 | 8.796 | 11.05 | 17.59 | 21.99 | 35.18 |
| | 50 | 1.414 | 2.827 | 5.498 | 6.283 | 10.99 | 13.82 | 21.99 | 27.48 | 43.98 |
| | 60 | 1.696 | 3.393 | 6.597 | 7.540 | 13.19 | 16.58 | 26.38 | 32.98 | 52.77 |
| | 70 | 1.979 | 3.958 | 7.697 | 8.796 | 15.39 | 19.35 | 30.78 | 38.48 | 61.57 |
| | 80 | 2.262 | 4.524 | 8.796 | 10.05 | 17.59 | 22.11 | 35.18 | 43.98 | 70.37 |
| | 90 | 2.545 | 5.089 | 9.896 | 11.31 | 19.79 | 24.88 | 39.58 | 49.48 | 79.16 |
| | 100 | 2.827 | 5.655 | 10.99 | 12.56 | 21.99 | 27.64 | 43.98 | 54.97 | 87.96 |
| | 110 | 3.110 | 6.220 | 12.09 | 13.82 | 24.19 | 30.41 | 48.38 | 60.47 | 96.762 |
| | 120 | 3.393 | 6.786 | 13.19 | 15.08 | 26.38 | 33.17 | 52.77 | 65.97 | 105.5 |
| | 130 | 3.676 | 7.351 | 14.29 | 16.33 | 28.58 | 35.94 | 57.17 | 71.47 | 114.3 |
| | 140 | 3.958 | 7.917 | 15.39 | 17.59 | 30.78 | 38.70 | 61.57 | 76.96 | 123.1 |
| | 150 | 4.241 | 8.482 | 16.49 | 18.85 | 32.98 | 41.46 | 65.97 | 82.46 | 131.9 |

**Table 9.** Angular Frequency ($f_a$). Based on the Temporal Frequency and Step Size.

| | | **Temporal Frequency ($f_t$) [Hz]** | | | | | | | | | |
| | | **3** | **6** | **7.5** | **12** | **15** | **24** | **30** | **48** | **60** | **95** |
|---|---|---|---|---|---|---|---|---|---|---|---|
| | 0.009 | 0.018 | 0.045 | 0.059 | 0.099 | 0.126 | 0.207 | 0.261 | 0.423 | 0.531 | 0.846 |
| | 0.018 | 0.036 | 0.090 | 0.117 | 0.198 | 0.252 | 0.414 | 0.522 | 0.846 | 1.062 | 1.692 |
| | 0.035 | 0.070 | 0.175 | 0.228 | 0.385 | 0.490 | 0.805 | 1.015 | 1.645 | 2.065 | 3.290 |
| | 0.040 | 0.080 | 0.200 | 0.260 | 0.440 | 0.560 | 0.920 | 1.160 | 1.880 | 2.360 | 3.760 |
| LiDAR's Step Size (°) | 0.070 | 0.140 | 0.350 | 0.455 | 0.770 | 0.980 | 1.610 | 2.030 | 3.290 | 4.130 | 6.580 |
| | 0.088 | 0.176 | 0.440 | 0.572 | 0.968 | 1.232 | 2.024 | 2.552 | 4.136 | 5.192 | 8.272 |
| | 0.140 | 0.280 | 0.700 | 0.910 | 1.540 | 1.960 | 3.220 | 4.060 | 6.580 | 8.265 | 13.16 |
| | 0.175 | 0.350 | 0.875 | 1.138 | 1.925 | 2.450 | 4.025 | 5.075 | 8.225 | 10.32 | 16.45 |
| | 0.280 | 0.560 | 1.400 | 1.820 | 3.080 | 3.920 | 6.440 | 8.120 | 13.16 | 16.52 | 26.32 |

**Table 10.** Minutes per Scan. Based on the Angular Frequency for a 360° scan.

| | | **Temporal Frequency ($f_t$) [Hz]** | | | | | | | | | |
| | | **3** | **6** | **7.5** | **12** | **15** | **24** | **30** | **48** | **60** | **95** |
|---|---|---|---|---|---|---|---|---|---|---|---|
| | 0.009 | 333.3 | 133.3 | 102.6 | 60.6 | 47.6 | 29.0 | 23.0 | 14.2 | 11.3 | 7.1 |
| | 0.018 | 166.7 | 66.7 | 51.3 | 30.3 | 23.8 | 14.5 | 11.5 | 7.1 | 5.6 | 3.5 |
| | 0.035 | 85.7 | 34.3 | 26.4 | 15.6 | 12.2 | 7.5 | 5.9 | 3.6 | 2.9 | 1.8 |
| | 0.040 | 75.0 | 30.0 | 23.1 | 13.6 | 10.7 | 6.5 | 5.2 | 3.2 | 2.5 | 1.6 |
| LiDAR's Step Size (°) | 0.070 | 42.9 | 17.1 | 13.2 | 7.8 | 6.1 | 3.7 | 3.0 | 1.8 | 1.5 | 0.9 |
| | 0.088 | 34.1 | 13.6 | 10.5 | 6.2 | 4.9 | 3.0 | 2.4 | 1.5 | 1.2 | 0.7 |
| | 0.140 | 21.4 | 8.6 | 6.6 | 3.9 | 3.1 | 1.9 | 1.5 | 0.9 | 0.7 | 0.5 |
| | 0.175 | 17.1 | 6.9 | 5.3 | 3.1 | 2.4 | 1.5 | 1.2 | 0.7 | 0.6 | 0.4 |
| | 0.280 | 10.7 | 4.3 | 3.3 | 1.9 | 1.5 | 0.9 | 0.7 | 0.5 | 0.4 | 0.2 |

In general, there are four categories for vibration amplitude/frequency that will dictate what is possible to capture via LiDAR, as shown in Table 11.

**Table 11.** Recommended LiDAR scanning goals based on the vibration frequency and amplitude being characterized.

| | Amplitude | Frequency | Goal |
|---|---|---|---|
| Low vibration amplitude | Less than 1.5 times the local accuracy | N.A. | Oversample with the goal of reducing noise within geometry estimates through averaging |
| Low vibration frequency | Greater than 1.5 times the local accuracy | Frequencies less than 0.5 Hz | Characterization of the vibration first mode and ensuring that the ripple patterns do not distort dimensional estimates |
| Moderate vibration frequency | Greater than 1.5 times the local accuracy | Frequencies within the 2–6 Hz bandwidth | Characterization of the vibration first mode and ensuring that the ripple patterns do not distort dimensional estimates |
| High vibration frequency | Greater than 1.5 times the local accuracy | Frequencies greater than 15 Hz | Oversample with the goal of reducing noise within geometry estimates through averaging |

**Author Contributions:** Conceptualization, A.T.-B.; Methodology, A.T.-B.; Formal analysis, A.T.-B.; Data curation, A.T.-B.; Writing—original draft, A.T.-B.; Writing—review & editing, J.G. and F.L.M.; Supervision, J.G. and F.L.M. All authors have read and agreed to the published version of the manuscript.

**Funding:** This research received no external funding.

**Data Availability Statement:** Not applicable.

**Conflicts of Interest:** The authors declare no conflict of interest.

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
