# Peer review of "Characterization of Operational Vibrations of Steel-Girder Highway Bridges via LiDAR"

_remotesensing, doi:10.3390/rs15041003_

Round 1

Reviewer 1 Report

Dear authors,

the work you presented is interesting and the paper is well organized. Moreover, the references cited are too few and I suggest inserting your research into a broader framework. I have some doubts about the structure of paragraph 3 where the reader can be confused by the presentation of the strategy adopted for the monitoring. It should be changed by moving the methodology to a new article section. Similarly, for paragraphs 4 and 5 where methodology and results are presented together. It should be better to reorganize the paper by providing a new section explaining the "theory" of your strategy. Conclusions should be improved also defining the results for the analyzed bridge. The last paragraph (recommendation) is incomplete; pay attention. 

Author Response

Thank you for showing interest and believing in the importance of this paper. We have carefully read and addressed all your comments. We have organized our response to each comment in the following table. We hope this revision improves the paper such that you now consider it worthy for publication.

We have included more references to bring up the state of the art of the technology. We also included more details on paragraph 3, to provide a better explanation of the technology. Additionally, paragraph 4 was expanded to provide more details of the technology, as mentioned before. On the recommendations, there was a gap within the description of “Step 3”. This was fixed.

Reviewer 2 Report

Dear authors

Unfortunately, the manuscript submitted for review in its current form has numerous shortcomings that prevent its publication.

  1. A review of the current state of knowledge in this field based on literature research is not sufficient. The cited literature contains only 12 items, of which as many as 9 are older than 5 years. Technologies of remote acquisition of information on engineering objects, including bridges, using laser scanners, but also other sensors, are extensively and in detail described in the literature of the subject in recent years. The lack of detailed literature research is a serious shortcoming in the field of rapidly developing measurement and data analysis technology.

2. The manuscript does not specify either the conditions for the measurements, or the accuracy specification of the measuring devices used, or even the names and models of the measuring devices used. Therefore, it is difficult to talk about verifying the accuracy or quality of the acquired measurement data and the possibility of verifying the presented results. From the very content of the scanning parameters given in the manuscript, one can conclude about the manufacturer of the scanner used, but this is definitely not enough.

3. No analysis of the accuracy of the measurements made by the laser scanner has been carried out. Both a priori and a posteriori. Therefore, it is impossible to verify whether the tested vibrations are significant in terms of measurement or whether they are within the limits of the measurement error. Anyway, the confirmation of the lack of a accyracy analysis of the measuring devices used and the sense of their use in given measurement conditions is Table 9 Minimum Amplitude of Vibration Detected. Local Accuracy (cm.) where values below 0.001 m occur. No terrestrial laser scanner rangefinder gives such distance measurement accuracy. Not to mention the impact on the accuracy of the position of the 3D point, errors in the measurement of horizontal and vertical angles, and errors caused by the angle of incidence of the laser beam on the measured surface. Of course, there are methods of data processing or surface estimation that allow to obtain higher accuracy than the nominal error of the position of a single point in the point cloud, but no information on this subject has been provided in the article. 

4. Determining the time frequency and angular frequency on the basis of the scanning frequency parameter set in the scanner (24 Hz - line 112) also raises doubts. What is the accuracy of the scanning frequency by the device and how stable is this scanning frequency throughout the measurement? How does the error in determining the scanning frequency affect the determination of the vibration parameters of the bridge structure? The ideal solution would be to have timestamps for scanner observations.

5. Figure 5: The descriptions for the vertical axis in point b overlap, in point c there are no descriptions at all.

6. Lines 155-156 errors in text editing, formula 6 incorrectly inserted between the lines.

7. Table 4, Table 5 and others - the units used are non-SI units.

8. Lines 326-328 are missing text

9. The information provided in chapter 7 Recommendations is unfortunately not universal and applies to one specific type of scanner, or scanners from a specific manufacturer - which, as already mentioned, have not been given. Not all devices have the ability to set the scan frequency, and this frequency is not always available in the device's technical documentation. Also, the angular resolutions available in the settings do not have to coincide with those given in the manuscript, or the sampling distance is defined at a given scanning distance.

To sum up, apart from minor editing errors that can be easily corrected, the manuscript lacks several key elements:

- there is no clear description of the conditions and assumptions for the implementation of field measurements;

- there is no description of the measuring equipment used with its accuracy parameters;

- there is no accuracy analysis and no description of the limitations of the measurement method resulting from it;

- the presented recommendations are not universal;

The study described as such is interesting and I hope that taking into account the above comments will allow you to prepare a publication that will not be objectionable.

Author Response

Thank you for showing interest and believing in the importance of this paper. We have carefully read and addressed all your comments. We have organized our response to each comment in the following table. We hope this revision improves the paper such that you now consider it worthy for publication.

Comment:

Action Taken:

Unfortunately, the manuscript submitted for review in its current form has numerous shortcomings that prevent its publication.

1. A review of the current state of knowledge in this field based on literature research is not sufficient. The cited literature contains only 12 items, of which as many as 9 are older than 5 years. Technologies of remote acquisition of information on engineering objects, including bridges, using laser scanners, but also other sensors, are extensively and in detail described in the literature of the subject in recent years. The lack of detailed literature research is a serious shortcoming in the field of rapidly developing measurement and data analysis technology.

Additional more recent references were included to provide this research a current perspective.

2. The manuscript does not specify either the conditions for the measurements, or the accuracy specification of the measuring devices used, or even the names and models of the measuring devices used. Therefore, it is difficult to talk about verifying the accuracy or quality of the acquired measurement data and the possibility of verifying the presented results. From the very content of the scanning parameters given in the manuscript, one can conclude about the manufacturer of the scanner used, but this is definitely not enough.

Detailed description of the technology used was included, as well as the scanner’s settings during data collection.

3. No analysis of the accuracy of the measurements made by the laser scanner has been carried out. Both a priori and a posteriori. Therefore, it is impossible to verify whether the tested vibrations are significant in terms of measurement or whether they are within the limits of the measurement error. Anyway, the confirmation of the lack of a accyracy analysis of the measuring devices used and the sense of their use in given measurement conditions is Table 9 Minimum Amplitude of Vibration Detected. Local Accuracy (cm.) where values below 0.001 m occur. No terrestrial laser scanner rangefinder gives such distance measurement accuracy. Not to mention the impact on the accuracy of the position of the 3D point, errors in the measurement of horizontal and vertical angles, and errors caused by the angle of incidence of the laser beam on the measured surface. Of course, there are methods of data processing or surface estimation that allow to obtain higher accuracy than the nominal error of the position of a single point in the point cloud, but no information on this subject has been provided in the article. 

The approximate measurement of the amplitudes of vibration observed in the field, and the ranging error or accuracy of the scanner were included in the text.

Thank you for your comments on Table 9. This table reflects the results gathered from the MatLab code analysis, where a vibrating plate and a stationary LiDAR was modeled to reflect the result of their interaction. Then, it is presented as an analysis of how much vibration can be detected based on the distance between the scanner and the vibrating object. This is not a reflection of field data results. 

4. Determining the time frequency and angular frequency on the basis of the scanning frequency parameter set in the scanner (24 Hz - line 112) also raises doubts. What is the accuracy of the scanning frequency by the device and how stable is this scanning frequency throughout the measurement? How does the error in determining the scanning frequency affect the determination of the vibration parameters of the bridge structure? The ideal solution would be to have timestamps for scanner observations.

Yes, we agree. Ideally, the meta data provided for each point should include time, additionally to the existent X, Y, Z coordinates. At this point, TLS do not record time for each data point collected.

This is why this paper included the reflection on the limitation of the TLS for this particular application.

5. Figure 5: The descriptions for the vertical axis in point b overlap, in point c there are no descriptions at all.

Thank you for your comment. The figure was modified.

6. Lines 155-156 errors in text editing, formula 6 incorrectly inserted between the lines.

Thank you for your comment. The text was modified.

7. Table 4, Table 5 and others - the units used are non-SI units.

Thank you for your comments. Units where corrected.

8. Lines 326-328 are missing text

Thank you for your comment. The text was modified.

9. The information provided in chapter 7 Recommendations is unfortunately not universal and applies to one specific type of scanner, or scanners from a specific manufacturer - which, as already mentioned, have not been given. Not all devices have the ability to set the scan frequency, and this frequency is not always available in the device's technical documentation. Also, the angular resolutions available in the settings do not have to coincide with those given in the manuscript, or the sampling distance is defined at a given scanning distance.

Thank you for your comment. Correct, this study was to evaluate the applicability of terrestrial laser scanners on detecting vibrations of highway bridges. The intention was also to reflect on the limitations of the technology in this particular presentation (TLS) other scanners might have other results and requirements.

The specifications of the scanner used were added to the text. Typical TLS used for bridge engineering applications are capable of varying the scanning frequency, also named resolution.

10. To sum up, apart from minor editing errors that can be easily corrected, the manuscript lacks several key elements:

10.a- there is no clear description of the conditions and assumptions for the implementation of field measurements;

Thank you for your comments. Additional explanation in the conclusions section was added.

10.b- there is no description of the measuring equipment used with its accuracy parameters;

This comment was addressed. Thank you again for your comment.

10.c- there is no accuracy analysis and no description of the limitations of the measurement method resulting from it;

Table 9 intends to present the levels of accuracy that the scanner could achieve based on distance to target and the scanner’s step size.

10.d- the presented recommendations are not universal;

Correct, the recommendations are to be used for TLS type of LiDAR.

The study described as such is interesting and I hope that taking into account the above comments will allow you to prepare a publication that will not be objectionable.

Thank you for highlighting the relevance of the research. We hope we have addressed all you comments on a way you now consider it ready for publication.

Reviewer 3 Report

The reviewer thanks the authors for their contribution. In general, the experimental and numerical work is appreciated, and the findings are useful information for future bridge applications. Nevertheless, the publication in the “Remote Sensing, MDPI” is not recommended unless the following suggestions are taken into account:

1) Introduction. The current state of knowledge relating to the manuscript topic has not been covered and clearly presented, and the authors’ contributions and findings are not emphasized. In this regard, the authors should make their effort to address these issues.

2) Section 4. The geometric and mechanical characteristics of the bridge have not clearly been illustrated. Please, provide more information and introduce figures with schemes of the bridge.

3) Please, report the layout of field experiments with locations of measuring points by LiDAR technology and conventional sensors. Please, specify.

4) The authors used conventional sensors. Please, specify the corresponding characteristics of the measurements, e.g., frequency (or the period) of the recording data etc. Moreover, range, sensitivity, resolution and accuracy of the conventional sensors should be underlined.

5) Section 4. Equation (6) is not necessary. Please, delete it.

6) Which finite element (FE) software has been used ? Please, cite the FE software in the text and references.

7) The FE analyses must be better explained with, particularly, details of the model used. It is not clear how the model of the bridge is composed (beam elements, plate and shell, etc., with the corresponding amounts and mesh sizes) and how the loading were applied. Which are the boundary conditions of the steel girders ? Moreover, have liner and/or nonlinear geometric analyses been performed ? Please, revise these parts and provide more information about the models.

8) Please, insert a table which lists the types of FE used, with the corresponding amounts within the models and mesh sizes.

9) The frequency measurements obtained from LiDAR technology, conventional sensors and FE modeling should be listed within a table with the corresponding comparison errors.

10) LiDAR technology also requires, for long-term monitoring of bridges over long distances, suitable environmental conditions, in situ visibility and intensive labor work. Is LiDAR suitable for long-term frequency measurements ? Please, specify.

11) Is LiDAR suitable for short-term vertical displacement measurements along bridges ? And what about long-term vertical displacement measurements ?

12) The frequency measurements from LiDAR are affected by the distance to target and incidence angle. That’s why conventional sensors still provide more accurate measurements in vibration frequency. Please, analyze advantages and disadvantages with conventional technologies which should be reported in the literature review (introduction). Also, refer to this issue through the following references which treat bridge laboratory and field experiments:

-  http://dx.doi.org/10.1016/j.engstruct.2016.08.006

-  https://doi.org/10.1016/j.istruc.2021.10.093

13) The further work, related to this study, should be mentioned at the end of the article. Please, specify.

Author Response

Thank you for showing interest and believing in the importance of this paper. We have carefully read and addressed all your comments. We have organized our response to each comment in the following table. We hope this revision improves the paper such that you now consider it worthy for publication.

Comment:

Action Taken:

The reviewer thanks the authors for their contribution. In general, the experimental and numerical work is appreciated, and the findings are useful information for future bridge applications. Nevertheless, the publication in the “Remote Sensing, MDPI” is not recommended unless the following suggestions are taken into account:

1) Introduction. The current state of knowledge relating to the manuscript topic has not been covered and clearly presented, and the authors’ contributions and findings are not emphasized. In this regard, the authors should make their effort to address these issues.

Thank you for your comment. The introduction has been modified in different paragraphs to address your comment.

2) Section 4. The geometric and mechanical characteristics of the bridge have not clearly been illustrated. Please, provide more information and introduce figures with schemes of the bridge.

Thank you for your comment. Additional information was been included in the text.

3) Please, report the layout of field experiments with locations of measuring points by LiDAR technology and conventional sensors. Please, specify.

Thank you for your comment. This information of provided in figure 4.

4) The authors used conventional sensors. Please, specify the corresponding characteristics of the measurements, e.g., frequency (or the period) of the recording data etc. Moreover, range, sensitivity, resolution and accuracy of the conventional sensors should be underlined.

Thank you for your comment. This information of provided in figure 5 and table 2.

5) Section 4. Equation (6) is not necessary. Please, delete it.

Equation 6 is presented to explain how the percent error between the frequency of the scanner and the frequency found in the dynamic test was calculated.

6) Which finite element (FE) software has been used ? Please, cite the FE software in the text and references.

The research did not use a finite element model. The images shown are from a numerical model generated in MATLAB.

7) The FE analyses must be better explained with, particularly, details of the model used. It is not clear how the model of the bridge is composed (beam elements, plate and shell, etc., with the corresponding amounts and mesh sizes) and how the loading were applied. Which are the boundary conditions of the steel girders ? Moreover, have liner and/or nonlinear geometric analyses been performed ? Please, revise these parts and provide more information about the models.

The research did not use a finite element model.

8) Please, insert a table which lists the types of FE used, with the corresponding amounts within the models and mesh sizes.

The research did not use a finite element model.

9) The frequency measurements obtained from LiDAR technology, conventional sensors and FE modeling should be listed within a table with the corresponding comparison errors.

Thank you for our comments. Additional information has been added to the text. The research did not use a finite element model.

10) LiDAR technology also requires, for long-term monitoring of bridges over long distances, suitable environmental conditions, in situ visibility and intensive labor work. Is LiDAR suitable for long-term frequency measurements ? Please, specify.

This is a great question. Terrestrial LiDAR are not suitable for long-term monitoring, since the scans would be performed under an asynchronous manner. Each scan would be independent, limiting the possibility to link the vibration data found. Each terrestrial LiDAR scan, as shown in this research, should be analyzed independently.

11) Is LiDAR suitable for short-term vertical displacement measurements along bridges ? And what about long-term vertical displacement measurements ?

These are great questions. If the load is applied and remains steady during the scanning process, then LiDAR could be able to detect the deformation if greater than the scanner’s accuracy. Then, you could compare the deformed shape to the undeformed shape (would have to scan without the load too).

TLSs are not recommended for long-term scanning under dynamic loading. This is concluded from this research.

12) The frequency measurements from LiDAR are affected by the distance to target and incidence angle. That’s why conventional sensors still provide more accurate measurements in vibration frequency. Please, analyze advantages and disadvantages with conventional technologies which should be reported in the literature review (introduction). Also, refer to this issue through the following references which treat bridge laboratory and field experiments:

-  http://dx.doi.org/10.1016/j.engstruct.2016.08.006

-  https://doi.org/10.1016/j.istruc.2021.10.093

Thank you for your comment. Additional information with the correspondent citations were included in the manuscript.

13) The further work, related to this study, should be mentioned at the end of the article. Please, specify.

Thank you. The correspondent citations were included.

Reviewer 4 Report

This manuscript explores the relationship between the scanning parameters of the LiDAR and the vibration parameters of the target, and puts forward a set of measurement methods with operational significance. But the paper needs to be revised to improve clarity and quality. The following issues need to be addressed:

1.       The literature review is not detailed enough and not relevant enough to the manuscript.

2.       The innovation point of this manuscript is not clear enough, and the proposed method is not compared with the existing advanced methods.

3.       In equation 3, the meanings of X1, X2, Y1 and Y2 are not clear. It is suggested to illustrate them in graphic form, The meanings of some symbols in equations 10 to 12 are not clear, such as LongLoc, Rev, etc. Please check the whole manuscript to avoid similar mistakes.

4.       Please check lines 155 to 158 and lines 326 to 332 for typographical errors.

Author Response

Thank you for showing interest and believing in the importance of this paper. We have carefully read and addressed all your comments. We have organized our response to each comment in the following table. We hope this revision improves the paper such that you now consider it worthy for publication.

Comments:

Action Taken:

This manuscript explores the relationship between the scanning parameters of the LiDAR and the vibration parameters of the target, and puts forward a set of measurement methods with operational significance. But the paper needs to be revised to improve clarity and quality. The following issues need to be addressed:

1.       The literature review is not detailed enough and not relevant enough to the manuscript.

Thank you for your comment. Additional references have been included.

2.       The innovation point of this manuscript is not clear enough, and the proposed method is not compared with the existing advanced methods.

Thank you for your comment. Additional information has been included to highlight the importance of the research.

3.       In equation 3, the meanings of X1, X2, Y1 and Y2 are not clear. It is suggested to illustrate them in graphic form, The meanings of some symbols in equations 10 to 12 are not clear, such as LongLoc, Rev, etc. Please check the whole manuscript to avoid similar mistakes.

A description has been added to the text preceding equation 3.

A description of the different parameters is resented in Table 5.

4.       Please check lines 155 to 158 and lines 326 to 332 for typographical errors.

Thank you for your comment. Errors have been fixed.

Round 2

Reviewer 1 Report

Dear authors,

the abstract should be more concise and some issues of the past form of the paper are still present (see the gap in the recommendation).  pay more attention, please.

Author Response

Dear authors,

the abstract should be more concise and some issues of the past form of the paper are still present (see the gap in the recommendation).  pay more attention, please.

Thank you for your comment. The abstract was improved to include additional explanation of the research.

The gap in the recommendation could be due to the inserted comments. We will correct before final submission.

Reviewer 2 Report

Dear authors

Thanks for the replies to the comments. The corrections made to the text are satisfactory.

However, I have 2 more comments:

1. SI units were used inconsistently. Further in the tables and in the text, imperial units appear, e.g.: table 5, lines 265 - 272. Please review the manuscript carefully and consequently convert the units to the SI system

2. Lines 361 - 365 are formatted incorrectly

Author Response

Dear authors

Thanks for the replies to the comments. The corrections made to the text are satisfactory.

However, I have 2 more comments:

1. SI units were used inconsistently. Further in the tables and in the text, imperial units appear, e.g.: table 5, lines 265 - 272. Please review the manuscript carefully and consequently convert the units to the SI system

Thank you. All units were revised and converted to SI units.

2. Lines 361 - 365 are formatted incorrectly

Thank you. The gap within the text appears when converting to pdf. We will make sure to address this issue before the final submission.

Reviewer 3 Report

The authors carried out the required revisions.

Author Response

The authors carried out the required revisions.

Thank you for accepting our previous answers to your comments.
